# Transition Temperature of Color Change in Thermochromic Systems and Its Description Using Sigmoidal Models

**DOI:** 10.3390/ma16237478

**Published:** 2023-12-02

**Authors:** Martina Viková, Michal Vik

**Affiliations:** Department of Material Engineering, Faculty of Textile Engineering, Technical University of Liberec, 46117 Liberec, Czech Republic; michal.vik@tul.cz

**Keywords:** thermochromism, color, sigmoidal function, textiles, temperature sensing

## Abstract

Background: Symmetric sigmoidal models with four parameters based on an idealized S/Z-shaped curve are commonly used to analyze the optical parameters of thermochromic materials. However, our experimental findings show that this approach leads to systematic errors involving the incorrect estimation of the transition temperature or the possibility of a virtual indication of the hysteresis nature of a reversible thermochromic change. For this reason, we sought to find a five-parameter model that would appropriately avoid this problem. Methods: Two commercial thermochromic pigments were used for the test and applied to a textile substrate at different concentrations. The optical properties were measured using reflectance spectrophotometry and then converted to Kubelka–Munk function values and colorimetric coordinates. The following statistics were used to assess the quality of the selected sigmoidal models: coefficient of determination, *R*^2^; adjusted coefficient of determination, *AR*^2^; root mean square error, *RMSE*; and Akaike Information Criterion, *AIC*. Results: The four-parameter models were compared with each other and with the five-parameter models using nested *F*-tests based on residual variance to obtain a statistical measure of superior performance. For all thermochromic color change data examined, the five-parameter models resulted in significantly better fitting. It could be shown that the five-parameter model showed significantly higher accuracy and precision in determining the transition temperature, like non-sigmoidal quantification methods. Conclusions: We concluded that the asymmetric five-parameter model is a valuable extension of the symmetric model in the investigation of thermochromic color changes, providing better parameter estimates and a new approach to investigating the mechanisms contributing to the asymmetry of the thermochromic curve.

## 1. Introduction

Chromic fabrics represent a specific area of indicators and sensory systems. Color change makes it easy to identify both the different stimuli and their intensity. In the case of thermochromic textiles, this involves temperature measurement or the identification of a specific pre-determined temperature. There are currently two systems that are based on a change of color depending on a specific temperature. The first of these systems is liquid crystals, which have the advantage of covering a temperature range of approximately 20 °C and are typically used as contact thermometers that can be placed on different surfaces [1,2,3,4,5,6,7]. The second system is usually a three-component composite system consisting of Leuco dye, a developer, and a suitable solvent [8,9,10,11,12]. This system can be reduced to a two-component system if the substance that acts as a solvent also acts as a developer [13]. Currently, a wide range of inks is available with a relatively wide temperature range where thermochromic change occurs, typically from −15 °C to 65 °C depending on the application [1,2,14,15,16,17,18]. In general, thermochromic materials can exhibit both irreversible and reversible color changes [18,19], and this paper focuses on the issue of reversible change. The thermochromic changes are schematically illustrated in Figure 1.

In the case of systems that exhibit irreversible color change, they are usually inorganic compounds [20,21], while in the case of reversible systems, they may be both inorganic and organic compounds or composite systems [22,23]. An accurate description of the color transition allows not only for the comparison of different thermochromic pigments but also for measuring possible changes in their behavior due to use. In the case of the use of composite thermochromic inks containing Leuco dye, a developer, and a suitable solvent, it is possible to find that the individual compositions differ from each other by examining the different average color change temperatures (the average temperature at which color change is observed) and the range of temperatures at which color change occurs [24,25].

If a thermochromic system is to be used as an indicator of reaching a certain temperature, it is preferable that the temperature range over which the color transition is achieved is as narrow as possible [26,27]. In addition, it is necessary to study the change in color response of the system during use. Most often, the number of cycles of thermochromic color change that the thermochromic ink can perform is studied. The influence of light, other factors, and effects of use on thermochromic systems are studied [28].

The color change in the case of composite thermochromic inks is caused by the phase change of the substance used as a solvent. This solvent is converted from a solid phase to a liquid phase, causing the developer and Leuco dye to separate and the resulting system to become colorless. In other words, it is possible to observe the change of state of the composite system from one state to another, while in published works usually only the record of the experimental points of the color change is evaluated, which has the shape of an S or a Z curve (this depends on the optical parameter measured), or the initial and final phases of the color change, where the corresponding temperatures are determined using a simple graphical procedure [14,29]. The principle of the graphic determination of the initial and final temperatures of the thermochromic color change consists of finding the intersections of a line through the linear part of the color transition data between two states, with the lines corresponding to the initial and final equilibrium of the measured color response of the thermochromic system. These changes can be analytically described using parametric functions such as third-order and higher-order polynomials, cotangential, sigmoidal, or logistical functions [30]. The advantage of parametric functions is the possibility of discovering the physical nature of the described phenomenon and comparing both the quality of the interleaving of experimental data and the study of possible errors associated with the appropriate determination of the parameters such as the initial, final, and mean temperature of the thermochromic transition [31]. As will be shown below, the most appropriate approach for this purpose is the use of sigmoidal models based on the original Sigmoidal Boltzmann Equation (SBE). This new approach to the analysis of thermochromic color change data also allows for reduced errors in the determination of the individual temperatures mentioned above when the analyzed data had an S-shape of color change versus a Z-shape and vice versa.

## 2. Materials and Methods

Two commercial thermochromic inks from Matsui International, Co., Kyoto, Japan, which are suitable for screen printing, were used to verify the usefulness of the considered sigmoidal function for estimating the thermochromic color curve. Gold Orange and Fast Blue, Type 37, for which Matsui states that the ink color is visible below 32 °C and the color disappears above 41 °C. Concentrations of thermochromic ink: 10, 50, 100, 150, and 300 g·kg^−1^ were used. A proven complex thickener system containing glycerin, Lukosan S (defoamer from Lučební závody a.s., Kolín, Czech Republic), Sokrat 492 (anionic binder from Chemické závody, Sokolov, Czech Republic), Acramin BA (binder from Bayer AG, Leverkusen, Germany, Lambicol L 90 (thickener from Lamberti S.p.A., Gallarate (VA) Italy), ammonia and water was used as printing paste.

### 2.1. Printing

The sample was printed on a Johannes Zimmer laboratory printing machine MINI-MD-R/F (Zimmer Maschinenbau GmbH, Klagenfurt, Austria) designed for screen printing. The printing was carried out using a flat screen and a roller squeegee. The squeegee was moved over the stencil using electromagnets placed under the printing blanket. The laboratory printing machine was set up to move the squeegee at a speed of 3 m·min^−1^ with a pressure of 3 N·cm^−1^. To minimize overprinting, only one squeegee stroke was carried out. Printing was followed by gradual drying of the samples on filter papers until they were dry to the touch. This was followed by fixation of the samples in an HS 122 hot-air oven at 80 °C for 5 min.

### 2.2. Measurement of Spectral and Color Parameters

The measurements were designed as isothermal measurements using two heated plates, one of which had a constant temperature of 20 °C using a Minichiller thermostat from Peter Huber Kältemaschinenbau SE, Offenburg, Germany and the second heated plate was gradually adjusted to temperatures of 30 to 45 °C in 1 °C steps using a Julabo F25-HE thermostat (JULABO GmbH, Seelbach, Germany). The actual reflectance measurement was carried out on a Datacolor D100 spectrophotometer (Lawrenceville, NJ, USA, aperture diameter 20 mm, mode: di:8° [32]), which was set to the vertical position, the sample was placed face down on the aperture measurement port and the corresponding heated plate was placed on top of the sample. First, the sample response was measured at 20° and then the thermal plate at a higher temperature was attached, and time measurements were made at intervals of 5 s to determine the kinetics of the color change and to determine the equilibrium of the color change corresponding to a condition where, on ten consecutive measurements, the deviation between measurements was no greater than 0.1 DE *. The result is the color response data for the two thermochromic inks tested and their different concentrations.

### 2.3. Spectral and Color Data Evaluation

The observed reflectance data were used to determine the Kubelka–Munk function values and tristimulus values *XYZ*, which represent CIE 1931 color space [33]:(1)$${\left(K/S\right)}_{\lambda }=\frac{{\left[1-\beta_{R}\left(\lambda \right)\right]}^{2}}{2\beta_{R}\left(\lambda \right)}$$
where *β_R_*(*λ*) is the reflected radiance factor [34], and *K* and *S* are Kubelka–Munk absorption and scattering coefficients.
(2)$$X=k\overset{830}{\underset{360}{\int }}\varepsilon \left(\lambda \right)\beta_{R}\left(\lambda \right)\bar{x}\left(\lambda \right)d\lambda$$
(3)$$Y=k\overset{830}{\underset{360}{\int }}\varepsilon \left(\lambda \right)\beta_{R}\left(\lambda \right)\bar{y}\left(\lambda \right)d\lambda$$
(4)$$Z=k\overset{830}{\underset{360}{\int }}\varepsilon \left(\lambda \right)\beta_{R}\left(\lambda \right)\bar{z}\left(\lambda \right)d\lambda$$
where *k* is the constant chosen such that *Y* = 100 for object, for which *β_R_*(*λ*) equals 1 for all wavelengths of selected wavelength interval; *ε*(*λ*) is spectral power of used light source, typically CIE (CIE—International Commission on Illumination (http://cie.co.at)) D65 illuminant and $\bar{x}\left(\lambda \right),\;\bar{y}\left(\lambda \right),\bar{z}\left(\lambda \right)$ are color matching functions of CIE standard observer, in this case, version 1931—2°. These functions are the numerical description of the chromatic response of the CIE standard observer—an average human observer in the range of 360–830 nm.

From these tristimulus values, color coordinates are computed in approximate uniform color space CIELAB [33] with use of following relations:(5)$$L^{*}=116f\left(Y/Y_{0}\right)-16$$
(6)$$a^{*}=500\left[f\left(X/X_{0}\right)-f\left(Y/Y_{0}\right)\right]$$
(7)$$b^{*}=200\left[f\left(Y/Y_{0}\right)-f\left(Z/Z_{0}\right)\right]$$
where function *f* equals to (τ/τ_0_)^1/3^ if (τ/τ_0_) > (24/116)^3^ or (841/108) (τ/τ_0_) *+* 16/116 if (τ/τ_0_) ≤ (24/116)^3^; τ are mentioned tristimulus values *X, Y, Z* of measured specimen; and τ_0_ is related to tristimulus values of ideal white diffuser-object for which *β_R_*(*λ*) equals 1 for all wavelengths of selected wavelength interval. The function *f* ensures that the transformation used from tristimulus values *X, Y, Z* corresponds as closely as possible to the nature of human color perception, with individual colors approximately evenly scaled.

The coordinates of the CIELAB color space form a rectangular coordinate system; however, a derived cylindrical coordinate system, which we refer to as CIELCH, is preferable in terms of visual color evaluation [33,35]. In this system, in addition to the lightness *L**, two other coordinates are used: the chroma *C** and the hue angle *h*°.
(8)$$C^{*}=\sqrt{{\left(a^{*}\right)}^{2}+{\left(b^{*}\right)}^{2}}$$
(9)h°=arctg(b*a*)

Using these coordinates, it is possible to consider the perception of individual differences using the human eye, so that an observer can usually distinguish the difference in brightness (10), chromaticity (11), and hue (12). The sample at the beginning of the measurement is usually used as a standard, and the change in color of the sample as the temperature increases or decreases is considered to be a corresponding batch related to the temperature used.
(10)$$∆L^{*}=L_{b}^{*}-L_{s}^{*}$$
(11)$$∆C^{*}=C_{b}^{*}-C_{s}^{*}$$
(12)∆H*=2Cb*.Cs*sin(∆h°2)
where subscript *b* refers to the batch and subscript *s* to the value relative to the standard.

The advantage of color difference evaluation is that it can be used in cases where the temperature indicator based on thermochromic inks will consist of two parts, one being color stable and the other consisting of a color changing thermochromic system. In this case, the two parts of the temperature indicator appear to be color identical below the temperature of thermochromic change and color different above the temperature of the color change. The human eye is more sensitive to these changes than to the color change of one uniform colored area [36].

In this work, the standard used was the measurement of the test sample of thermochromic ink at 20 °C before the start of the warm-up phase of the test. The measurements that corresponded to the equilibrium color of the sample at the selected test temperature were then used as the batch, as discussed in the previous section.

### 2.4. Mathematical Modeling

The thermochromic color change in thermochromic inks containing dye and developer is characterized using a sigmoid transition along with an inflection point, as is the case with many other phase transitions [37,38,39,40]. A frequently used function in this case is the sigmoidal function proposed by Boltzmann (1879) [37], which has the following univariate binary form: *y* = 1/(1 + *e^x^*). In our case, we used a modified version that allows us to determine the inflection point of the color transition and temperature distances (the difference between the temperature at which the color change of the thermochromic system begins and the temperature at which it ends). This form is called Sigmoidal Boltzmann Equation (SBE):(13)$$y=y_{b}+\frac{\left(y_{f}-y_{b}\right)}{A+e^{\frac{c-x}{\alpha }}}$$
where *y_b_* and *y_f_* are equilibrium values of the dependent variable (*K/S*, Δ*L**, Δ*C**, Δ*H** and Δ*E**) before and after transition; *c* is critical value of stimuli *x*; in our case transition temperature *T*_T_, which is also abscissa of inflection point.

*A* is a parameter that adjusts the position of the critical value relative to the geometric location of the transition between two states. If *A* = 1, the critical value is in the middle of the transition, and it is true that *y_c_* = (*y_b_* + *y_f_*)/2. In cases where *A* is not equal to 1 but not very different from 1 the critical value of the transition is located near the mid [point]. Because in the case of calculating the fit of the experimental data with the model (13) with an alternative value of the parameter *A*, the calculations proved to be unstable in some parameters, a fixed value of *A* = 1 was chosen, when the calculations were stable, and the statistical criteria confirmed the validity of the model.

Parameter α describes the properties of the color transition slope and identifies a possible discontinuity in the monitored process. If α is close to zero, then the first derivative of model (13) approaches infinity and discontinuous transition is observed. Parameter α is also a curvature parameter that indicates over what temperature range the color of the thermochromic material changes. In the case of sensing materials, it is important that this range is as narrow as possible. In the case that the transition temperature is not significantly different from the temperature determined as the center of the color transition between the two states of the thermochromic sensor, Equation (13) is applicable for such determinations. The final version of SBE is the following for the Kubelka–Munk function, with a similar solution for the colorimetric parameters Δ*L**, Δ*C**, Δ*H** and Δ*E**.
(14)$$\left(K/S\right)={\left(K/S\right)}_{b}+\frac{{\left(K/S\right)}_{f}-{\left(K/S\right)}_{b}}{1+e^{\frac{c-T}{\alpha }}}$$

The transition point generally assumes that 50% of the thermochromic ink has undergone color transformation. However, in addition to the transit temperature, the temperatures of interest are those showing 10% and 90% transformation of the thermochromic ink, which are usually found at the beginning and end of the approximately linear part of SBE, where the curved parts of the function leading asymptotically to the limiting values end or begin. To address these cases, we can make the following modification of Equation (14), where the temperature of 10% or 90% thermochromic change is denoted as *T*_R_.
(15)$$e^{\frac{c-T_{R}}{\alpha }}=\frac{{\left(K/S\right)}_{f}-{\left(K/S\right)}_{b}}{{\left(K/S\right)}_{R}-{\left(K/S\right)}_{b}}-1$$

Using simple modifications of Equation (17) we obtain:(16)$$\frac{c-T_{R}}{\alpha }=ln\left[\frac{{\left(K/S\right)}_{f}-{\left(K/S\right)}_{b}}{{\left(K/S\right)}_{R}-{\left(K/S\right)}_{b}}-1\right]$$
(17)$$T_{R}=c-\alpha \left\{ln\left[\frac{{\left(K/S\right)}_{f}-{\left(K/S\right)}_{b}}{{\left(K/S\right)}_{R}-{\left(K/S\right)}_{b}}-1\right]\right\}$$

Equation (19) has a solution only under the condition that [(KS)f−(KS)b(KS)R−(KS)b−1]>0.

## 3. Results and Discussion

The reflectance values of the measured samples were converted to the Kubelka–Munk function (1) and colorimetric parameters (2–12) according to the above equations. The colorimetric parameters and the CIE *Yxy* and CIELAB coordinates are shown graphically in the graphs in Figure 2. These graphs show that during the heating of the thermochromic fabrics, the observed samples are gradually decolorized, which is manifested as a decrease in the excitation purity for the CIE *xy* diagram and in the chromaticity in plane *a* b** for the CIELAB color space. At the same time, there is an increase in brightness *Y* and lightness *L**. In the case of the *a* b** plane of the CIELAB color space, it is also seen that the decoloring process follows a curve and not a straight line as one might assume. The reason for this behavior is that the concentration dependence of the sample position in the color space is affected in real colorants both by the substrate itself, which is not ideally achromatic, and by the change in the spectral reflectance waveform, where, after a certain minimum reflectance is reached, with a further increase in concentration there is a decrease in reflectance around the dominant wavelength of the colorant, which ultimately results in a decrease in chroma and a certain hue shift. If we express the values of the Kubelka–Munk function (*K*/*S*) as a function of temperature, we obtain a typical sigmoidal curve for thermochromic color change with well resolved individual concentrations of the tested thermochromic samples, as can be seen in the plots in Figure 3.

At the same time, one can see the fit of the experimental points using a symmetric SBE. Table 1 and Table 2 show the results for SBE (14) for both tested inks, individual concentrations, and estimates of the temperatures at which 50%, 10% and 90% decolorization of the tested inks occurs. The estimates of temperatures *T*_R10_ and *T*_R90_ are then used to determine the basic temperature interval in which a significant thermochromic color change occurs. Its width, denoted as temperature distance Δ*T*, then indicates the suitability of the thermochromic system in terms of its sensory application. As already mentioned, the aim is to keep the temperature distance in which the color change occurs as narrow as possible. Both inks approximately fall into the manufacturer’s declared category referred to as 37, where the temperature range can be 32 °C up to 41 °C. (https://matsui-color.com/ourproducts/specialty-products/chromicolor, accessed 25 November 2023). However, their temperature distances are differently sized and differently dependent on concentration, i.e., Δ*T* depends both on the ink used and, above all, on its concentration.

From the graph in Figure 4A, the thermochromic ink ‘Gold Orange’ has a significantly higher dependence of Δ*T* on temperature (essentially an exponential decrease approaching 2.5 °C) compared to the ink ‘Fast Blue’. From the plot in Figure 4B, it is also evident that the temperature distance Δ*T* is the inverse of the slope coefficient α. Expressing the thermal distance Δ*T* from Equation (17) for the case that Δ*T* = *T*_R90_ − *T*_R10_, we obtain Equation (18):(18)$$∆T=2\alpha \left|ln\left[\frac{{\left(K/S\right)}_{f}-{\left(K/S\right)}_{b}}{{\left(K/S\right)}_{R}-{\left(K/S\right)}_{b}}-1\right]\right|$$

Equation (18) has the following solution, since the difference (*K*/*S*)*_f_ −* (*K*/*S)_b_* represents 100% of the color change of the thermochromic ink. If we consider the difference of the measured values at 10% conversion of the thermochromic ink as the situation where the difference (*K*/*S*)*_R_ −* (*K*/*S*)*_b_* represents 90% and at 90% conversion as the difference when (*K*/*S*)*_R_ −* (*K*/*S*)*_b_* represents 10%, then we obtain the following solution of Equation (18):$$\mathrm{Conversion}\;10\%:\;∆T=2\alpha \left|ln\left[\frac{100}{90}-1\right]\right|=2\alpha \times \left|2.1860178\right|=2\alpha \times 2.1860178$$
$$\mathrm{Conversion}\;90\%:\;∆T=2\alpha \left|ln\left[\frac{100}{10}-1\right]\right|=2\alpha \times \left|-2.1860178\right|=2\alpha \times 2.1860178$$

And Equation (18) can be rewritten into a simple form:(19)$$∆T=2\alpha .\ln\left(\frac{10}{9}-1\right)$$

It was mentioned that the solution of Equations (14)*–*(20) given for the Kubelka–Munk function (*K*/*S*) can also be applied on colorimetric color parameters (Δ*L**, Δ*C**, Δ*H**, and Δ*E**). Their advantage is that they can be measured both with simple colorimeters and, for example, with digital cameras. Moreover, this evaluation is closer to the perception of the human eye, which is important in terms of the application of thermochromic systems as simple visual indicators of a certain temperature. In addition to the asymmetric progression of the individual difference curves, we can also note the difference in the contribution of the partial differences (Δ*L**, Δ*C**, and Δ*H**) to the overall color difference between the compared inks, Figure 5 and Figure 6. In the case of the Fast Blue ink, we see that the contribution of the difference in lightness Δ*L** has the highest influence on the total color difference Δ*E**, then for the Gold Orange ink we see that the main component of the total color difference Δ*E** is the difference in chroma Δ*C**.

An interesting pattern in the measured data can also be seen in the hue difference Δ*H**, which reaches a maximum in the transition temperature area and then returns to the minimum difference. Also, in the case of the Fast Blue ink, we can see that the proportion of the hue difference has a relatively small effect on the total color difference Δ*E**. From this, we can conclude that the evaluation of the transit temperature from the progression of the hue difference dependence will be the least effective and, therefore, in the next section, we will concentrate on the analysis of the progression of the remaining differences, i.e., Δ*E**, Δ*L**, and Δ*C**.

### 3.1. Asymmetric Models of Sigmoidal Fit

Over the years, several asymmetric sigmoidal functions have been developed. The first tested equation was published by Gompertz [41] and is an example of a mathematical model for time series; however, it can also be used for the case of general sigmoidal trends including the color change of the thermochromic ink developer tested in this work. All the following equations can also be written for Δ*L** and Δ*C**.
(20)$$∆E^{*}=∆E_{b}^{*}+\left(∆E_{f}^{*}-∆E_{b}^{*}\right)e^{-e^{\left(\frac{c-T}{\alpha }\right)}}$$

The Gompertz equation is a special case of a generalized logistical function that is also known as the Richards [42] function, which was included as a second asymmetric model:(21)∆E*=∆Eb*+(∆Ef*−∆Eb*)[1+νe(c−Tα)]1ν
where ν is constant of asymmetry.

The next pair of equations are the Double Sigmoidal Boltzmann Equations DSBE [43]. The first of these, denoted as DSBE1, has the following form:(22)$$∆E^{*}=∆E_{b}^{*}+∆E_{f}^{*}\left[\frac{p}{1+e^{\left(\frac{c_{A}-T}{\alpha }\right)}}+\frac{1-p}{1+e^{\left(\frac{c_{B}-T}{\beta }\right)}}\right]$$
where β is the second curvature parameter and C_A_/C_B_ are the critical stimulus values (the temperatures related to the first and second part of the curvature of the transition function).

The second Double Sigmoidal Boltzmann Equation, denoted as DSBE2 [44], is written as follows:(23)$$∆E^{*}=\frac{∆E_{b}^{*}}{1+e^{\left(\frac{c_{A}-T}{\alpha }\right)}}+\frac{∆E_{f}^{*}-∆E_{b}^{*}}{1+e^{\left(\frac{c_{B}-T}{\beta }\right)}}$$

The last model tested was BARO5, which is a five-parameter model extended from the four-parameter SBE model with the addition of a second curvature parameter. This model is used to assess arterial baroreflex using vasoactive agents allowing investigators to collect pairs of data over a wide range of blood pressures and reflex responses [45].
(24)$$∆E^{*}=∆E_{b}^{*}+\frac{∆E_{f}^{*}-∆E_{b}^{*}}{1+fe^{\left(\frac{c-T}{\alpha }\right)}+\left(1-f\right)e^{\left(\frac{c-T}{\beta }\right)}}$$
where *f* defines a logistic weighting function varying smoothly between 0 and 1 centered around *c* (V50), and if *α* and *β* are of the same sign, the mean curvature *f* is given by:(25)$$f=\frac{1}{1+e^{\left(-\frac{c-T}{\gamma }\right)}}$$
(26)$$\gamma =\frac{2\alpha \beta }{\left|\alpha +\beta \right|}$$

### 3.2. Assessment of the Best Model Using Statistical Tests

The most common way of comparing the quality of equations modelling a particular dependence is to use the coefficient of determination *R*^2^ [46], which is calculated according to the following equation:(27)$$R^{2}=1-\frac{RSS}{TSS}$$
with *RSS* being the residual sum of squares $RSS=\sum_{i=1}^{n}{\left(y_{i}-{\hat{y}}_{i}\right)}^{2}$ and *TSS* the total sum of squares $TSS=\sum_{i=1}^{n}{\left(y_{i}-{\bar{y}}_{i}\right)}^{2}$, where *n* represents number of observations.

Ideally, *RSS* = 0 and the coefficient of determination reaches 1. Except for Gompertz’s model, the other tested models of the asymmetric sigmoidal function are nested models, i.e., they are extensions of the original four parameters of SBE (13) using an additional parameter or a doubling. As a result, the *F*-test, which quantifies the relative decrease in the sum of squares when moving from a simpler model to a more general model, can be used to select the best fitting model.
(28)F=(RSS4−RSS5)(df4−df5)RSS5df5
where *RSS*_4_ is the residual sum of squares of a four-parameter fit; *RSS*_5_ is the residual sum of squares of a five-parameter fit; and *df*_4_ and *df*_5_ are the degrees of freedom for a four- and five-parameter fit.

Another way to test the relative quality of the individual models is the Akaike Information Criterion (*AIC*). This criterion tests the relative quality of a model, i.e., in relation to other models. The lower the *AIC*, the better the model fits the data and the higher its ranking [47]. For the case of small data sets, which is also the case for thermochromic color change measurements with a 1 °C step, the recommended *AIC* form is as follows:(29)$$AIC=n\ln\left(\frac{RSS_{i}}{n}\right)+2k_{i}+\frac{2k_{i}\left(k_{i}+1\right)}{n-k_{i}-1},\;n\neq k_{i}+1$$
where *k_i_* is the number of model parameters plus one.

The *AIC* primarily rewards improvement in the goodness of fit to the experimental data, while to account for increased uncertainty, the *AIC* also includes a penalty that is an increasing function of the number of parameters.

Like the *AIC*, the Adjusted Coefficient of Determination *AR*^2^ also handicaps the increase in the number of parameters in the model when assessing the quality of the fit to the experimental data, and its formula is as follows:(30)$$AR^{2}=1-\left(1-R^{2}\right)\;\frac{n-1}{n-p-1}$$
where *p* is the number of parameters.

The last criterion used in this paper is the root-mean-squared-error *RMSE*, which allows comparing different models using absolute *RMSE* values, where the smaller the *RMSE*, the better the tested model fit.
(31)$$AR^{2}=1-\left(1-R^{2}\right)\;\frac{n-1}{n-p-1}$$

All asymmetric models were tested against the SBE (except for the Gompertz model), i.e., in a nested F-test, the asymmetric model match was considered to be an alternative hypothesis (H_1_) to the null hypothesis (H_0_), i.e., the SBE match. Thus, the best performing model is indicated in bold in the table. The tables below also allow for a comparison of all the models using all the statistical criteria mentioned.

For readability, only the results for the total color difference ΔE* and one concentration of thermochromic inks are shown here, the other results are available from the authors upon request. All tests and parameter estimates were performed using GraphPad Prism 10 statistical software for macOS (version 10.1.1) and were also cross-checked using OriginPro 2018 on the Microsoft Windows platform (version b.9.5.1.195).

One of the reasons for testing asymmetric models of thermochromic color change in this work is the difference in transition temperature estimates using spectrophotometric data and colorimetric evaluation. Note in Table 3 and Table 4 that the transition temperature estimate *T*_T_ by the symmetric SBE model is 38.0 °C for the total color difference ΔE* and the Fast Blue thermochromic ink, which is 1.3 °C higher than the *T*_T_ estimate using the Kubelka–Munk function (Table 1). In the case of the Gold Orange (Table 2) ink, the difference between the two estimates of *T*_T_ is lower, but here again the *T*_T_ estimate using the total color difference ΔE* is 0.8 °C higher. This is a problem, of course, because in the case of one original data set of the reflected radiance factor, we expect the estimate of the transition temperature to be the same for both the Kubelka–Munk and the colorimetric data, since both are based on this data set.

Table 3 and Table 4 show that the DSBE fit using the Double Sigmoidal Boltzmann Equations was the most effective (the first in order is marked in bold). In the case of the Fast Blue thermochromic ink, this was the DSBE2 model, whereas in the case of the Gold Orange thermochromic ink it was the DSBE1 model. In the case of the DSBE2 and the Fast Blue ink, we see that this model was the best performing in terms of *AIC* and *AR*^2^. In terms of comparison using *R*^2^ and *RSME,* this model was second in order. For this reason, a comparison using the *F*-test was made between this model, DSBE2, and BARO5.

The difference between the *T*_T_ estimates decreases to 1 °C for the Fast Blue thermochromic ink and to 0.3 °C for the Gold Orange thermochromic ink. Thus, the question remains why the colorimetric data lead to a higher temperature estimate than the Kubelka–Munk function. In the plots in Figure 7, we see that the fit to the experimental data is tighter in the case of the asymmetric sigmoidal models than in the case of the SBE. Thus, the same comparison needs to be made in the case of the fit for the Kubelka–Munk function (*K*/*S*) data. The results of the evaluation of each model for both pigments are summarized in Table 5 and Table 6. It is evident that the use of asymmetric models results in a shift in the *T*_T_ estimation compared to the SBE. For the Fast Blue thermochromic ink, model DSBE1 was evaluated as the most effective for (*K*/*S*), in contrast to the total color difference data where it was model DSBE2. In the case of both inks, we can see that the data for model DSBE2 is not filled because the calculation of the estimation of some parameters was unstable.

From this it can be concluded that this model is suitable for the case of S-curves, not for Z-curves such as the curve (*K*/*S*). For the Gold Orange thermochromic ink, BARO5 was judged to be the best in case (*K*/*S*), but the difference in performance between this model and DSBE1 is very small. Similarly, we can evaluate the ranking of the individual models for the Fast Blue thermochromic ink and the overall color difference data, where DSBE2 was evaluated as the best performing model, but the results of DSBE1 are similar. The fit of the experimental data to the Kubelka–Munk function and the color difference Δ*Ε** using the DSBE1 model is shown in the plots in Figure 8. The fit of the experimental data with the DSBE1 model is tighter than that of the four-parameter SBE model. This makes up for the difference between the adaptation of the spectral data (*K*/*S*) and the colorimetric data Δ*Ε**. On the other hand, it should be considered that the double sigmoid models provide estimates of two temperatures, which may differ significantly from each other, as demonstrated in the case of the Fast Blue ink, where the difference is 2.4 °C. This is due to the nature of the data, which for the curve (*K*/*S*) leads to two displaced areas for the calculation of the inflection point, as documented in the graph in Figure 8.

## 4. Conclusions

The results discussed showed that the symmetric Sigmoidal Boltzmann Equation SBE may be a source of underestimating the position of the inflection point in case of descending Z-curves. Conversely, in the case of ascending S-type sigmoidal curves, the position of the inflection point is overestimated. As a result, there is a difference in the estimates of the transit temperature *T*_T_ thermochromic color change determined using spectrophotometric and colorimetric data. Among the set of asymmetric sigmoidal models, the Double Sigmoidal Boltzmann Equation DSBE1 was the best performing for the evaluation of the Kubelka–Munk function data and the overall color difference. The resulting difference between the transition temperatures determined using the two mentioned data sets decreased from 1.3 °C to 0.4 °C for the Fast Blue thermochromic ink and from 0.8 °C to 0.2 °C for the Gold Orange thermochromic ink. A certain drawback of the DSBE1 method is the estimation of two critical temperatures, which can be influenced by local fluctuations in the measured data, resulting in a secondary inflection region of the curve that corresponds to the data. In such cases, the BARO5 model, which unifies the calculation of the critical temperature of the thermochromic color change for both asymmetric parts of the sigmoidal transition, seems preferable.

## Figures and Tables

**Figure 1 materials-16-07478-f001:**
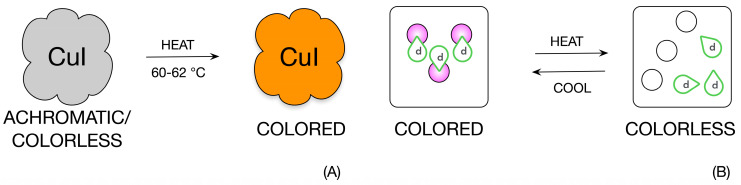
Schematic representation of thermochromic change. Part (**A**) of the figure is an illustration of irreversible thermochromic change and part (**B**) demonstrates reversible thermochromic change. The reversible thermochromic change in this case is an illustration of composite thermochromic inks, where *d* stands for the developer and the circle stands for the Leuco dye itself.

**Figure 2 materials-16-07478-f002:**
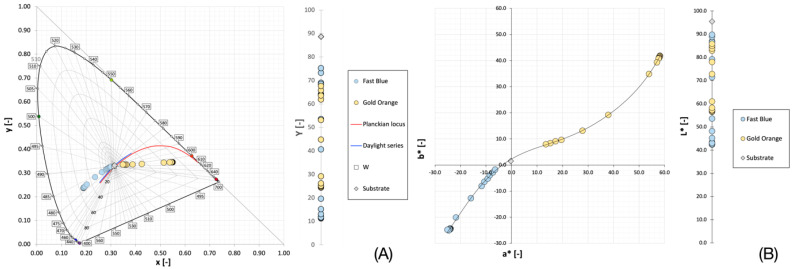
Representation of the positions of individual thermochromic samples (blue and orange) at different temperatures in the CIE Y, x, y color space (**A**) and in the CIELAB color space (**B**). The concentration of both samples was 150 g·kg^−1^. The vertical axes represent a change in brightness (Y) and lightness (L*), respectively, of both colors of thermochromic samples on CIE diagram x, y (**A**) and in CIELAB color space (**B**). Concentration of both samples was 150 g·kg^−1^.

**Figure 3 materials-16-07478-f003:**
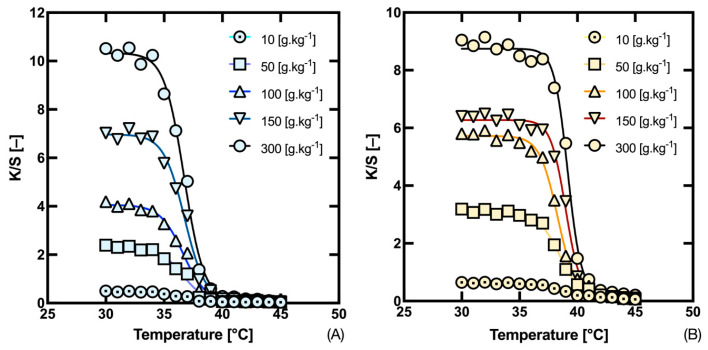
(*K*/*S*) transition as a function of temperature for inks (**A**) Fast Blue; (**B**) Gold Orange.

**Figure 4 materials-16-07478-f004:**
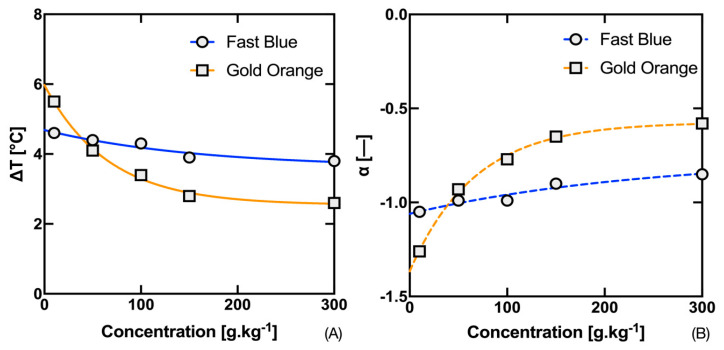
Dependence of temperature distance Δ*T* (**A**) and slope coefficient α (**B**) on the concentration of Fast Blue and Gold Orange thermochromic inks.

**Figure 5 materials-16-07478-f005:**
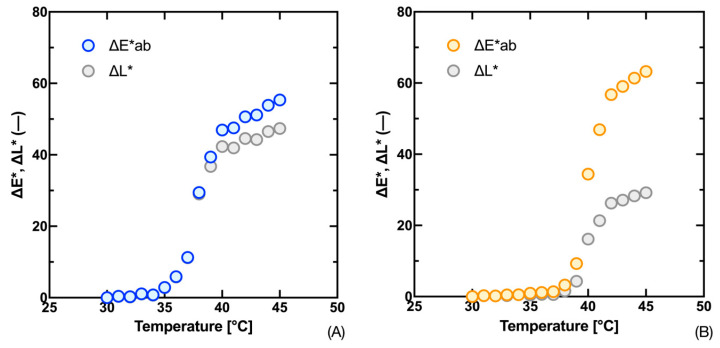
Measured values of total color difference Δ*E** and lightness difference Δ*L** in relation to sample temperature for 300 g·kg^−1^ Fast Blue (**A**) and 300 g·kg^−1^ Gold Orange (**B**) inks.

**Figure 6 materials-16-07478-f006:**
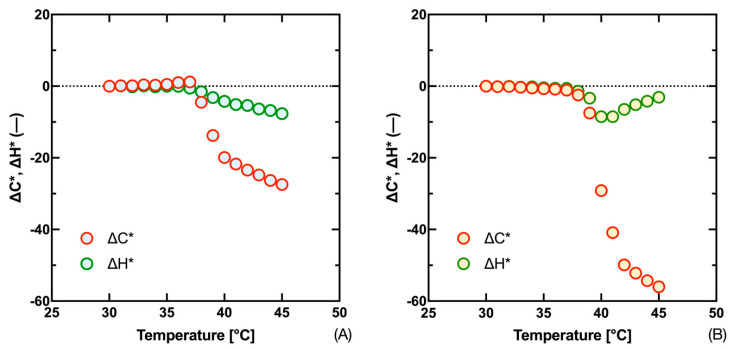
Measured values of chroma difference Δ*C** and hue difference Δ*H** in relation to sample temperature for 300 g·kg^−1^ Fast Blue (**A**) and 300 g·kg^−1^ Gold Orange (**B**) inks.

**Figure 7 materials-16-07478-f007:**
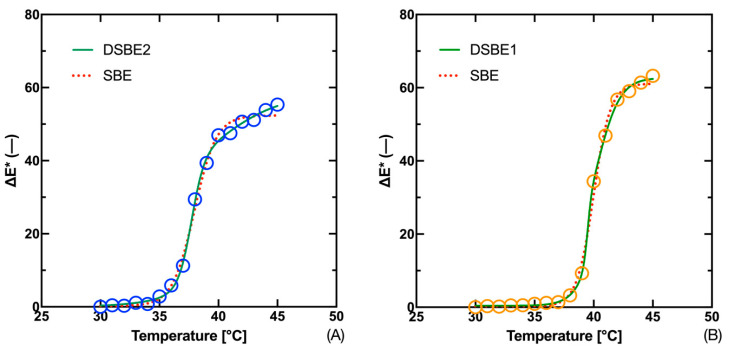
Fit of temperature dependence of total color difference Δ*E** for 300 g·kg^−1^ Fast Blue (**A**) and 300 g·kg^−1^ Gold Orange (**B**) inks.

**Figure 8 materials-16-07478-f008:**
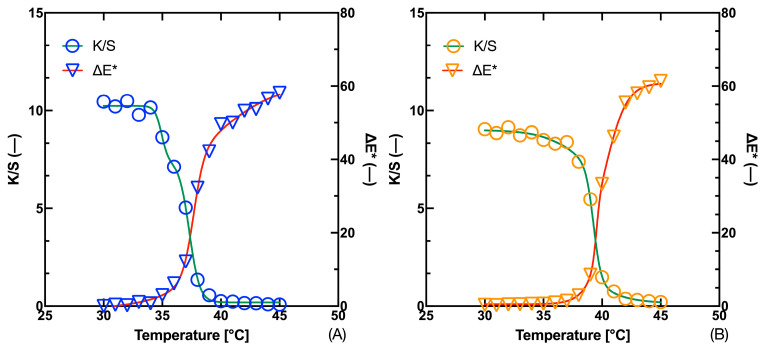
Fit of temperature dependence of (*K*/*S*) and total color difference Δ*E** for 300 g·kg^−1^ Fast Blue (**A**) and 300 g·kg^−1^ Gold Orange (**B**) inks.

**Table 1 materials-16-07478-t001:** Estimated parameters of Equations (14) and (17) for Fast Blue thermochromic ink.

Concentration [g·kg^−1^]	(*K*/*S*)*_b_* [—]	(*K*/S)_f_ [—]	*T*_T_ [°C]	*α* [—]	*T*_R10_ [°C]	*T*_R90_ [°C]	Δ*T* [°C]
10	0.040	0.484	36.5	−1.05	34.2	38.8	4.6
50	0.024	2.329	36.6	−0.99	34.4	38.7	4.4
100	0.019	4.058	36.6	−0.99	34.5	38.8	4.3
150	0.045	6.967	36.8	−0.90	34.8	38.7	3.9
300	0.072	10.330	36.7	−0.85	34.8	38.6	3.8

**Table 2 materials-16-07478-t002:** Estimated parameters of Equations (14) and (17) for Gold Orange thermochromic ink.

Concentration [g·kg^−1^]	(*K*/*S*)*_b_* [—]	(*K*/*S*)_f_ [—]	*T*_T_ [°C]	*α* [—]	*T*_R10_ [°C]	*T*_R90_ [°C]	Δ*T* [°C]
10	0.081	0.636	38.8	−1.26	36.0	41.6	5.5
50	0.125	3.108	38.4	−0.93	36.4	40.5	4.1
100	0.175	5.721	38.3	−0.77	36.6	40.0	3.4
150	0.093	6.274	39.0	−0.65	37.6	40.4	2.8
300	0.254	8.746	39.2	−0.58	37.9	40.4	2.6

**Table 3 materials-16-07478-t003:** Estimated *T*_T_ and evaluation criteria of tested models for 300 g·kg^−1^ Fast Blue thermochromic ink and total color difference Δ*E**.

Tested Equation	*T*_T_ [°C]	*F* (DFn, DFd)	*AIC* [—]	*R^2^* [—]	*AR*^2^ [—]	*RMSE* [—]
SBE (14)	38.0	—	29.25	0.9956	0.9945	1.562
Gompertz (22)	37.8	—	39.86	0.9915	0.9894	2.177
Richards (23)	37.6	1.82 (1, 11) H_0_	32.13	0.9962	0.9949	1.447
DSBE1 * (24)	37.7 (40.6)	9.09 (3, 9) H_1_	27.52	0.9989	0.9982	0.778
DSBE2 * (25)	37.7 (40.5)	13.86 (2, 10) H_1_	20.01	0.9988	0.9983	0.805
BARO5 (26)	37.9	7.19 (1,11) H_1_	26.54	0.9973	0.9964	1.215

* Two critical temperatures *c*_A_ and *c*_B_ are calculated in the DSBE equations, therefore, the second temperature is given in brackets.

**Table 4 materials-16-07478-t004:** Estimated *T*_T_ and evaluation criteria of tested models for 300 g·kg^−1^ Gold Orange thermochromic ink and total color difference Δ*E**.

Tested Equation	*T*_T_ [°C]	*F* (DFn, DFd)	*AIC* [—]	*R*^2^ [—]	*AR*^2^ [—]	*RMSE* [—]
SBE (14)	40.00	—	29.12	0.9966	0.9958	1.556
Gompertz (22)	39.6	—	19.08	0.9982	0.9977	1.137
Richards (23)	39.6	9.58 (1,11) H_1_	24.43	0.9982	0.9975	1.138
DSBE1 * (24)	39.5 (40.4)	27.62 (3, 9) H_1_	12.52	0.9997	0.9994	0.487
DSBE2 * (25)	38.4 (38.9)	4.60 (2, 10) H_1_	30.67	0.9982	0.9974	1.123
BARO5 (26)	39.9	41.42 (1, 11) H_1_	9.466	0.9993	0.9990	0.713

* Two critical temperatures *c*_A_ and *c*_B_ are calculated in the DSBE equations, therefore, second for the Gold Orange thermochromic ink, the DSBE1 was evaluated as the best performing model in terms of *R*^2^, *AR*^2^ and *RSME*, including the *F*-test against the DSBE2 and BARO5 models. At the same time, we can see that in both cases, the transition temperature estimates for thermochromic inks are lower than when using SBE.

**Table 5 materials-16-07478-t005:** Estimated *T*_T_ and evaluation criteria of tested models for 300 g·kg^−1^ Fast Blue thermochromic ink and Kubelka–Munk function (*K*/*S*).

Tested Equation	*T*_T_ [°C]	*F* (DFn, DFd)	*AIC* [—]	*R*^2^ [—]	*AR*^2^ [—]	*RMSE* [—]
SBE (14)	36.7	—	−23.09	0.9957	0.9947	0.305
Gompertz (22)	36.8	—	−11.16	0.9910	0.9888	0.442
Richards (23)	37.2	7.07 (1, 11) H_1_	−25.69	0.9974	0.9965	0.238
DSBE1 * (24)	37.3 (34.9)	7.25 (3, 9) H_1_	−22.17	0.9988	0.9979	0.165
DSBE2 * (25)	-	-	-	-	-	-
BARO5 (26)	36.8	6.41 (1, 11) H_1_	−25.10	0.9973	0.9963	0.242

* Two critical temperatures *c*_A_ and *c*_B_ are calculated in the DSBE equations, therefore the second temperature is given in brackets.

**Table 6 materials-16-07478-t006:** Estimated *T*_T_ and evaluation criteria of tested models for 300 g·kg^−1^ Gold Orange thermochromic ink and Kubelka–Munk function (*K*/*S*).

Tested Equation	*T*_T_ [°C]	*F* (DFn, DFd)	*AIC* [—]	*R*^2^ [—]	*AR*^2^ [—]	*RMSE* [—]
SBE (14)	39.2	—	−28.71	0.9958	0.9948	0.255
Gompertz (22)	39.2	—	−17.85	0.9918	0.9897	0.359
Richards (23)	39.5	4.273 (1, 11) H_0_	−28.63	0.9970	0.9959	0.217
DSBE1 * (24)	39.3 (38.35)	8.117 (3, 9) H_1_	−29.10	0.9989	0.9981	0.133
DSBE2 * (25)	-	-	-	-	-	-
BARO5 (26)	39.2	10.60 (1, 11) H_1_	−34.18	0.9979	0.9971	0.182

* The parameters of the DSBE1 model are not shown because their calculation was unstable.

## Data Availability

All experimental data are available from the authors.

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
