# Peer review of "Transition Temperature of Color Change in Thermochromic Systems and Its Description Using Sigmoidal Models"

_materials, 2023, doi:10.3390/ma16237478_

Round 1

Reviewer 1 Report

Comments and Suggestions for Authors

Manuscript Title : Transition temperature of color change in thermochromic systems and its description by sigmoidal models

Manuscript ID :    materials-2729015

The manuscript studied the modelization of thermochromic color changes with  asymmetric five-parameter model as an extension of the symmetric one. A fifth parameter was added to 4-parameter symmetric model to avoid incorrect estimation of the transit temperature or the possibility of a virtual indication of the hysteretic nature of the reversible thermochromic change. 

The paper offers a well structured and comprehensive point of view on the field of thermochromic materials. The methodology is well described and the litterature analysis well conceived. The conclusions are consistent and the arguments address in an appropriate way the topic. The references are appropriate and connected with the argumentation. Tables and figures are all right. In addition, the paper falls well with the journal topics.

I recommend before publishing this paper in materials journal, the authors could make these minor improvements:

-        Delete subtitles like (1) Background , (2) Methods  from the “Abstract section”.

-        The novelty of the research work is not very clear in “introduction section”. Try to focus on the limitation of the symmetric model and the need for an adjusting fifth parameter.

-        Page 9, Eqs (24) and (25), give the definition of “β” parameter .

-        Page 11, Line 327. Add if possible an explanation to such phenomena !

Reviewer 2 Report

Comments and Suggestions for Authors The manuscript presented six different models to describe the temperature transition of thermochromic pigments. The manuscript presented interesting fitting functions, applied to experimentally determined spectrophotometric measurements based on commercial pigments. It is logically presented into sections, however some of the equations can be moved to the methodology as detailed in the attached pdf document. Moreover the literature is missing a more detailed introduction of the range of temperatures and semiconductor thermochromic materials with appropriate temperature transitions for wider exploitation of the materials. Detailed line-by-line comments can be found in the attached document and should be carefully addressed before the manuscript is considered.

Comments on the Quality of English Language

Minor editing of English language required

Reviewer 3 Report

Comments and Suggestions for Authors

The work concerns an attempt to improve existing models of color change under the influence of temperature (thermochromic change). The presented problem seems to be technologically important.

The caption under Fig.1 should be more precise, i.e. a clear indication of which transition is reversible and which is not, e.g.Figs.1a... and Fig.1b.....

Eq.1 describe the coefficient bR

Line 113-  ?Ì…(?), ?Ì…(?), ?Ì…(?) are color matching functions - provide a more precise physical interpretation of these functions

Eq.2,3,4- what exactly do the XYZ trichromatic values represent? (what represents X, Y and Z).

Eq. 5,6 and 7 - how they were derived, can provide the source literature

Line 117-118. The interpretation of the function f and its parameters should be explained more clearly.

Eq. 8, 9 - how they were derived, can provide the source literature

Eq.13. What is the basis for using this modified form of Sigmoidal Boltzmann Equation (SBE)?

The theoretical background presented is confusing and not very clear. There is no good description of the variables used, making the text not very clear. This should be more clear for the reader, which will allow them to analyze the idea that is the basis of this article.

Comments on the Quality of English Language

The language of work is understandable

Round 2

Reviewer 2 Report

Comments and Suggestions for Authors

The authors provided revisions to the manuscript, however it is not clear where did these revisions address the comments of the review and how. Providing line by line responses, can address this concern. Moreover, the position of the methodology is reasonable and can remain as is, however the introductory literature and available temperature ranges with alternative thermochromic materials are still not clear.

Author Response

Thank you for this comment and we attach the following addressable explanation of the individual changes in the form of a file containing the original text and the text after revision 1 and 2, respectively. The temperature range in which thermochromic inks are used is generally conducive to low-melting solvents such as higher fatty acids, their derivatives, and the like. For the temperature ranges, we have also added references to the two books on chromium materials where several general details are given, including a reference to the article [14] where the colorimetric parameters of thermochromic inks are discussed.

Reviewer 3 Report

Comments and Suggestions for Authors

Thank you very much for the changes and relevant comments

Comments on the Quality of English Language

The language of work is sufficiently understandable. However, there is always space for improvement

Author Response

We agree with the reviewer that it is possible to improve our English a little. We have made a few corrections to the text for better readability and had the text edited by native speakers within our university's proofreading system. The language corrections made are shown in red in the attached file.
